Only two out of five articles by New Zealand researchers are free-to-access: a multiple API study of access, citations, cost of Article Processing Charges (APC), and the potential to increase the proportion of open access

White Richard K.A. richard.white@otago.ac.nz 1
Angelo Anton 2
Fitchett Deborah 3
Fraser Moira 2
Hayes Luqman 4
Howie Jessica 5
Richardson Emma 6
White Bruce 7
1 Vice-Chancellor’s Office, University of Otago , Otago , New Zealand
2 Library, University of Canterbury , Christchurch , New Zealand
3 Learning, Teaching and Library, Lincoln University , Lincoln , New Zealand
4 Library, Auckland University of Technology , Auckland , New Zealand
5 Library, University of Waikato , Hamilton , New Zealand
6 Libraries and Learning Services, University of Auckland , Auckland , New Zealand
7 Office of the Provost, Massey University , Palmerston North , New Zealand
Stern David
Electronic publication date: 2021 May 26
Publication date: 2021
Volume: 9
Electronic Location ID: e11417
Received 2020 Aug 4; Accepted 2021 Apr 16
Copyright: ©2021 White et al.
Copyright year: 2021
Copyright holder: White et al.
License: This is an open access article distributed under the terms of the Creative Commons Attribution License, which permits unrestricted use, distribution, reproduction and adaptation in any medium and for any purpose provided that it is properly attributed. For attribution, the original author(s), title, publication source (PeerJ) and either DOI or URL of the article must be cited.
License URL: https://creativecommons.org/licenses/by/4.0/

Keywords: Open Access, API, author’s rights, Article Processing Charges (APC), Hybrid Open Access, Citation advantage, Green Open Access, Repositories, Open Access policy, Scholarly communication

Funding: The authors received no funding for this work.

==============================
We studied journal articles published by researchers at all eight New Zealand universities in 2017 to determine how many were freely accessible on the web. We wrote software code to harvest data from multiple sources, code that we now share to enable others to reproduce our work on their own sample set. In May 2019, we ran our code to determine which of the 2017 articles were open at that time and by what method; where those articles would have incurred an Article Processing Charge (APC) we calculated the cost if those charges had been paid. Where articles were not freely available we determined whether the policies of publishers in each case would have allowed deposit in a non-commercial repository (Green open access). We also examined citation rates for different types of access. We found that, of our 2017 sample set, about two out of every five articles were freely accessible without payment or subscription (41%). Where research was explicitly said to be funded by New Zealand’s major research funding agencies, the proportion was slightly higher at 45%. Where open articles would have incurred an APC we estimated an average cost per article of USD1,682 (for publications where all articles require an APC, that is, Gold open access) and USD2,558 (where APC payment is optional, Hybrid open access) at a total estimated cost of USD1.45m. Of the paid options, Gold is by far more common for New Zealand researchers (82% Gold, 18% Hybrid). In terms of citations, our analysis aligned with previous studies that suggest a correlation between publications being freely accessible and, on balance, slightly higher rates of citation. This is not seen across all types of open access, however, with Diamond OA achieving the lowest rates. Where articles were not freely accessible we found that a very large majority of them (88% or 3089 publications) could have been legally deposited in an institutional repository. Similarly, only in a very small number of cases had a version deposited in the repository of a New Zealand university made the difference between the publication being freely accessible or not (125 publications). Given that most New Zealand researchers support research being open, there is clearly a large gap between belief and practice in New Zealand’s research ecosystem.

Introduction

Researchers seek to change the world and writers seek to be read, but for many years a dysfunctional scholarly publishing system has walled off most published research findings from the majority of its potential readership. Since the transition from print to electronic publishing began in the late 1990s, various initiatives explored the potential for this digital transformation to make research literature more accessible to the public. University libraries, concerned about continuing growth in journal subscription costs, hoped an open access system would provide a more affordable alternative. At the same time they sought to advance the mission of their host institutions to create social capital through the promulgation of quality peer-reviewed research.

Three major developments in the early 2000s set the scene for the current open access environment:

• The growth of “Gold open access” journals, funded by Article Processing Charges (APCs) rather than by subscriptions.

• The adoption of “Hybrid open access” options by subscription journals, making individual papers openly accessible through the payment of APCs.

• The growth of institutional and subject-specific repositories, providing an alternative route of “Green open access” to individual papers without publication charges.

Since then there has been considerable interest in the potential of open access to contribute to universities’ own goals as a result of supporting broader society to access research outputs. This includes a growing understanding that, as a result of their enhanced availability, openly accessible papers are likely to be cited at a higher rate than those behind paywalls.

Unfortunately, open access has not produced the anticipated reduction in costs. Subscription costs of research journals continue to rise while APCs for Gold and Hybrid journals add another cost to university budgets (Guédon, Kramer & Laakso, 2019). Furthermore, whereas subscription costs were centralised within library budgets, APC charges are paid from a variety of sources, including departmental budgets and external research funds, which makes them less visible and harder to manage (Monaghan et al., 2020). Moreover libraries have had limited success in encouraging researchers to deposit copies of their work in institutional repositories. In New Zealand, this is despite all universities having an institutional repository. The emergence of transformative agreements has also been a significant development. These agreements vary, but at their core they share the goal of shifting from subscription-based reading to contractually-based publishing. Although transformative agreements are becoming more common, they are still only a factor for a small number of institutions and their development is outside of the scope of this research.

New Zealand has no specific guidance from government or major research funding agencies on open access publishing or centralised support to pay APCs. While government has established an open access framework that applies to government agencies, this does not extend to the university sector (New Zealand Government, 2014). A recent government consultation document on research strategy raised the possibility that a co-ordinated approach in the research sector could be of benefit. To date, New Zealand’s major funding agencies have not enforced requirements on research projects they fund to release research outputs or data with open licences. Some New Zealand universities have adopted open access policies and/or guidelines following the Green open access pathway (Wikipedia contributors, 2020), as there is no government funding to enable Gold open access. Three universities operate small APC funds to support researchers to publish open access when this will achieve certain goals.

In 2019 the Council for New Zealand University Libraries (CONZUL) established a project team with representatives from seven of the eight New Zealand universities to research the current environment of open access in New Zealand. A major stream of this project sought to establish how much of our universities’ research outputs were open access. While other tools provide a figure for the proportion of research that is open, we wanted to extend our understanding to determine:

• how much our researchers might be spending on Article Processing Charges (APCs) on top of what libraries pay in subscriptions;

• how much of our work could be freely accessible via self-archiving but is not; and

• the relationship between openness and citations or other measures of impact.

This paper reports on the findings of this project and makes our method and software code available to others to create their own sets of data and their own analyses.

This paper focuses on one element of the CONZUL Open Access Project. The wider project produced a full report (Fraser et al., 2019) examining the wider open access environment in New Zealand and an infographic designed to communicate its findings in a readily digestible format.

Literature Review

As the prevalence of open access publication of research results has increased over the years (Abediyarandi & Mayr, 2019; Archambault et al., 2014; Archambault et al., 2013; Gargouri et al., 2010; Laakso et al., 2011; Maddi, 2019; Martín-Martín et al., 2018; Piwowar et al., 2018; Wang et al., 2018), so too has the ability to gain insight into its nature and development. However, this has occurred alongside increasing complexity in the way open access levels are measured, and the resulting literature is methodologically diverse. A recent publication highlighted the need for critical reflection on the methods employed to measure open access development in order to address regional and political inequity (Huang et al., 2020). This literature review presents a brief overview of the main methodological approaches and relevant results.

Perhaps the most influential study in recent years was carried out by Piwowar et al. (2018). In their review of the literature, they note the paucity of studies between 2014 and the time of writing, however further large-scale studies quickly followed (Robinson-Garcia, Costas & Leeuwen, 2020; Huang et al., 2020). As more automated research on open access becomes possible through Application Programming Interfaces (APIs) and enhanced indexing, sample sizes have increased (Piwowar et al., 2018). How open access development is measured depends on a number of factors, including scope and source of data. Different methods are used according to the aims of the research. Some studies focus on a given country (Abediyarandi & Mayr, 2019; Bosman & Kramer, 2019; Holmberg et al., 2019; Mikki, Gjesdal & Strømme, 2018; Piryani, Dua & Singh, 2019; Pölönen et al., 2019; Sivertsen et al., 2019) open access type (Wang et al., 2018) or funder (Kirkman, 2018). Others aim for a global overview (Archambault et al., 2014; Laakso et al., 2011; Martín-Martín et al., 2018; Piwowar et al., 2018; Robinson-Garcia, Costas & Leeuwen, 2020; Wang et al., 2018).

Because of this diversity, it is difficult to draw comparisons between results. Most recent studies point to an overall open access rate of between 45 and 55% (Bosman & Kramer, 2019; Martín-Martín et al., 2018; Piwowar et al., 2018; Pölönen et al., 2019). This is significant because an open access rate of 50% is posited as a “tipping point” by some (Archambault et al., 2013). Where open access rate is calculated as total of the scholarly record or over an extended period, this figure drops dramatically—Piwowar et al. (2018) estimate the total percent of scholarly record at 28%, Maddi (2019) at 31%. Piwowar, Priem & Orr (2019) hypothesize a rise to 44% by 2025. Where wider sources are included, such as Academic Social Networks (ASNs) or Google Scholar the open access rate rises (Martín-Martín et al., 2018; Nazim & Zia, 2019). Unsurprisingly, older studies report lower rates (Bjork et al., 2010; Gargouri et al., 2010), reinforcing the findings of many scholars that open access is on the rise (Abediyarandi & Mayr, 2019; Archambault et al., 2014; Archambault et al., 2013; Gargouri et al., 2010; Laakso et al., 2011; Maddi, 2019; Martín-Martín et al., 2018; Piwowar et al., 2018; Wang et al., 2018). The growth of Gold open access in particular has been noted (Archambault et al., 2014; Martín-Martín et al., 2018; Piryani, Dua & Singh, 2019; Pölönen et al., 2019). Despite this, the literature clearly shows that open access development varies by discipline (Bjork et al., 2010; Bosman & Kramer, 2019; Maddi, 2019; Martín-Martín et al., 2018; Piryani, Dua & Singh, 2019; Robinson-Garcia, Costas & Leeuwen, 2020; Sivertsen et al., 2019) and country (Archambault et al., 2013; Maddi, 2019; Martín-Martín et al., 2018; Robinson-Garcia, Costas & Leeuwen, 2020; Sivertsen et al., 2019; Torres-Salinas, Robinson-Garcia & Moed, 2019; Wang et al., 2018).

It is important to note that all of these results can only be viewed as accurate at a given point of time. Both Archambault et al. (2013) and Robinson-Garcia, Costas & Leeuwen (2020) point out that freely accessible research does not always adhere to the tenets of “open access”. Open access is fluid in nature—closed articles can become open after embargoes, repositories can be backfilled and publications may be open access but only at the discretion of the publisher or on limited terms This is manifest in the fact that Piwowar et al. (2018) introduced the Bronze category, defined as “made free-to-read on the publisher website, without an explicitly open license”. They noted that “It is also not clear if Bronze articles are temporarily or permanently available to read for free” (p.6). In fact, there is limited consensus on the definition of all open access types and even on the line between open access and non-open access. Similarly, the results of any study are also closely linked to its scope and source data—many studies use Web of Science or Scopus, which are known to under-represent certain areas of literature (Martín-Martín et al., 2018). As automation becomes more central to open access research, results become limited to those publications with a DOI, further excluding some important categories of research. Robinson-Garcia, Costas & Leeuwen (2019) argued that sources of open access status such as Unpaywall also need to be better understood in order to fairly represent open access rates.

The funding of open access through article processing charges (APCs) is another matter of high concern, although there is limited consensus in the literature around how these costs are to be estimated. A journal is classified as Gold if all articles are immediately open and APCs for these titles are recorded in the Directory of Open Access Journals (DOAJ), while for Hybrid journals the articles are paywalled unless an APC is charged. One method of estimating the cost of APCs to institutions is by examining financial records (Jahn & Tullney, 2016; Pinfield, Salter & Bath, 2017; Solomon & Björk, 2016) which aims to capture the actual amounts paid or by reviewing institutional agreements with publishers (Lovén, 2019). The other main approach is capturing the advertised prices from DOAJ or publisher websites (Björk & Solomon, 2015; Matthias, 2018; Morrison et al., 2016; Solomon & Björk, 2016).

Citation advantage is another topic that has been hotly debated in the literature. Research almost always finds a positive correlation between open access and citation rate (Archambault et al., 2014; Copiello, 2019; McCabe & Snyder, 2014; Mikki, Gjesdal & Strømme, 2018; Ottaviani, 2016; Piwowar et al., 2018; Piwowar, Priem & Orr, 2019; Wang et al., 2015). However confounding factors cast considerable uncertainty over direct causation (Gaulé & Maystre, 2011; Torres-Salinas, Robinson-Garcia & Moed, 2019). It is also clear that citation advantage is not distributed evenly across all disciplines (Holmberg et al., 2019) or types of open access (Mikki, Gjesdal & Strømme, 2018; Piwowar et al., 2018). In fact, several studies have found a citation dis advantage for Gold open access (Archambault et al., 2014; Archambault et al., 2013; Piwowar et al., 2018; Torres-Salinas, Robinson-Garcia & Moed, 2019). The way citation advantage (or lack thereof) is measured can have a considerable influence on results, leading some scholars to use normalised figures such as Category Normalised Citation Impact (Torres-Salinas, Robinson-Garcia & Moed, 2019) or Average Relative Citation (Archambault et al., 2016) rather than total citations. Others argue that quality bias from self-selection (i.e., researchers select open access for higher quality work) inflate the apparent citation advantage (Torres-Salinas, Robinson-Garcia & Moed, 2019). However, some authors have found that a citation advantage exists despite confounding factors, albeit at a lower rate (Gargouri et al., 2010; McCabe & Snyder, 2014; Ottaviani, 2016). Attention, as measured by views, downloads and altmetrics, are similarly positively affected (Adie, 2014; Holmberg et al., 2019; Wang et al., 2015) and Wang et al. (2015) found that downloads for open access publications were sustained for longer periods of time than non-open access. On balance, the literature largely confirms the open access citation advantage but the magnitude and reasons for this remain unclear.

Materials & Methods

The CONZUL project team developed software that used Digital Object Identifiers (DOIs) to establish publications’ open access status, APC price, and ability to be self-archived.

Our work depended on many open API services, the most integral being Unpaywall. As such our definition of ‘open’ in this study largely aligns with that of Unpaywall, including Bronze as initially proposed by Piwowar et al. (2018). Thus the openness of an article in our study is defined very broadly: “OA articles are free to read online, either on the publisher website or in an OA repository.” Unpaywall does “not harvest from sources of dubious legality like ResearchGate or Sci-Hub” (Unpaywall, 2021). Table 1 shows the categories we used and an associated definition.

Unpaywall uses a hierarchy to determine a single status for each paper. Priority is given to those statuses which imply immutability, specifically through publication in a Gold journal or through the payment of an APC in a Hybrid journal. For Gold journals no distinction is made between those that charge APCs and those that do not, following the definition of Gold as any form of open publication regardless of business model, as offered by Suber (2012). In our study, we were particularly interested in distinguishing between paid and unpaid forms of OA. Therefore, where the Directory of Open Access Journals (DOAJ) showed a Gold journal does not charge APCs, we re-categorised these as Diamond. This means we use the term ‘Gold’ in this paper exclusively to mean publication in an OA-only journal where an APC is charged. As already noted, Unpaywall introduced the Bronze status for papers openly available from the publishers but without an explicit license. Perhaps unfortunately, given questions around the persistence of Bronze open access, this status was given a higher priority than Green, which was reserved for papers openly accessible from repositories rather than from publishers. In this paper we use Green with the specific meaning that a publication has been made available in a reputable repository and that this is the only open version. Thus, while there may be overlap between categories, each publication is only given a single type of OA: for example, a paper may have been published in a Hybrid journal and deposited by a researcher in a repository; since the published version is ‘better’ according to the Unpaywall hierarchy, this is classified as Hybrid not Green. The status Closed is defined as papers that are not openly available in any form.

Table 1 Open access type definition.

Definitions of types of access used in this analysis.

Type of open access	Definition	
Gold OA	Published version is immediate OA. APC charged.	
Hybrid OA	Publication is subscription-based. APC can be paid to make individual articles OA.	
Bronze OA	Currently free to read on publisher’s site but licence not clear.	
Green OA	Only accessible in a reputable repository (i.e., academic social networking sites are not included). Publisher’s version is paywalled.	
Closed	Published version is paywalled.	
Diamond OA	Published version is immediate OA. Derived from Unpaywall giving the status ‘Gold’ but where DOAJ shows no APC charged.	

Piwowar et al. (2018) found that under the version of the API running in 2018 77% of all papers identified as open by a manual accuracy check were correctly identified as such and that 96% of the papers identified as open by the API were in fact open. The main variations are likely to occur from repository copies not being identified and Bronze papers reverting to Closed status over time. The Unpaywall API is widely used and provides robust comparability with other studies.

Overlaying the access dimension is the question of authorship. The number of authors of a published research article can range from one (sole authorship) to several thousand (project participation). Multiple authorship is a significant issue when we attempt to link published research to institutions and countries, particularly when there are no established norms for allocating divisions of responsibility. Where there are, say, 200 authors in a research group the fact that one of them is employed at University A tells us very little about the behaviour and performance of that institution, although a productive project may end up crediting it with numerous publications on the basis of participation by this single team member. This may be an insoluble problem for affiliation-based bibliometric research but in a project like the present one it is advisable not to ignore it. One means of creating a “strong link” between a paper and an institution is through the “corresponding author” who takes overall responsibility for the publication process. While this is often the first-named author, this is not universal.

For our purposes, we limited our sample set to journal articles with a Digital Object Identifier (DOI) published in 2017 that included at least one author affiliated with a New Zealand university. This provided a comprehensive dataset representing a large proportion of the research outputs of all eight universities in the country. Although we were carrying out the work in 2019 we chose to use 2017 as our sample set because, firstly, the research outputs were more likely to have passed the date for embargo set by publishers for self-archiving (one of our key interests) and, secondly, citation counts would be more mature than for more recent research.

DOIs for 2017 journal articles were gathered from each university, then amalgamated into a single file of more than 12,600 journal articles. If there was a local corresponding author at any university for a given article then it was designated as having a New Zealand corresponding author. During the course of the project we found that a small percentage of articles with large numbers of authors and large numbers of citations skewed the data so articles with more than 20 authors were excluded on the grounds that they had a tenuous connection to the New Zealand University that had submitted the DOI. This reduced the sample size to 12,016. These were fed into the Program.

The Program

At the heart of our work was the ‘Program’, written in Python. One of our primary aims in publishing this paper is to share the code for the Program for others to use as well as detailing the results of our own work. The code is available here: https://github.com/bruce-white-mass/conzul-oa-project.

A set of DOIs can be submitted to the Program, which uses a number of APIs to produce a set of results, whether for a single department, an institution, a discipline, a country (as in our case) or any other parameter. For our project, having compiled our list of DOIs as described above, we fed them into the Program using a Comma Separated Value file (.CSV). For each article the following information was obtained from a range of sources as shown in Table 2.

Errors are reported by the Program where information could not be found, for example if a DOI was not recognised.

DOIs were obtained from the individual universities. It was then possible to “chain” the data gathering. For example, Unpaywall provided ISSNs which were then submitted via API requests to Sherpa/Romeo to capture data on publisher allowances for the use of publications in institutional repositories. ISSNs were also used to capture data on APCs for individual journals.

However, not all the data used by the Program was accessible through APIs. Crossref was an excellent source of information for authors, even when these numbered in the thousands, but provides very limited data on author affiliations. On the other hand, Web of Science and Scopus provide detailed author-affiliation data, including identifying corresponding authors, but this needed to be output manually as CSV files for subsequent access by the program. A similar process was followed with APC data.

While this paper is focused on the national picture for New Zealand, for those who may be interested in utilising our code on their own DOIs we note that author affiliation data is included in the output. Therefore results can also be broken down to analyse subsets at the level of individual institutions.

The process described here is also presented in Fig. 1.

Results

The Program was run on 30 May 2019. The output was analysed and the following information extracted:

• the overall percentage of open and closed papers both for all authors and the subset of New Zealand corresponding authors;

• the total percentage of papers in each of the access categories: Closed, Gold, Hybrid, Bronze, Green, Diamond (note that the “best version” is reported so there was no overlap between categories.);

• the total percentage of papers available through repositories (note that because an article can be published and in a repository there is some overlap with the other categories);

• the total percentage of open and closed papers funded by major New Zealand agencies;

• the total cost for Gold and Hybrid papers if all APCs had been charged as advertised;

• the total cost of APCs as advertised if they had been paid on papers available in repositories;

• the total number of closed papers that could be made open as Author Accepted Manuscripts (AAM) as deduced from allowances recorded in Sherpa/Romeo;

• the total cost of APCs as advertised if these papers were made open in Hybrid mode.

Overall proportion of open v closed articles

Overall 59% of all the articles in our sample set were only available behind a subscription paywall (see Table 3).

Table 2 Information gathered by the Program and corresponding data sources.

Multiple sources of information were used in this analysis, listed here.

Information	Source	
Metadata: author(s), title, journal, ISSN, etc.	Unpaywall API	
Open or closed	Unpaywall API	
Type of access (Gold, Hybrid, Green, Closed, etc.)	Unpaywall API	
Reprint/corresponding author	Web of Science and Scopus (CSV file)	
Funders	Web of Science and Scopus (CSV file)	
URLs of all repository versions	Unpaywall API	
Type of ‘best open version’: published, preprint, postprint	Unpaywall API	
APC/No APC	DOAJ (CSV file) GitHub site of Lisa Matthias (Freie Universität Berlin) https://github.com/lmatthia/publisher-oa-portfolios	
Journal and publisher archiving rules and embargo periods	Sherpa/Romeo API	
Citations	Crossref API	

Figure 1 Flowchart showing process of data gathering, data sources and filtering applied to create dataset.

This flowchart can be used to filter our dataset to match the subsets of information used in our analysis.

When we performed the same analysis of those articles where the corresponding author was affiliated with a New Zealand university (as opposed to any of the authors being from a New Zealand university) we found the proportion of open articles was significantly less (see Table 4).

The proportion of open here reduced to 34%, meaning only 1 in 3 articles from 2017 where the corresponding author was affiliated with a New Zealand university was freely accessible.

Articles by type of access

As seen in Table 5, APC-incurring Gold open access comprises the largest proportion of open articles in our sample (35% of open articles and 14% of all articles). Green is next (26% of open articles or 10% of all articles), closely followed by Bronze. Hybrid is a significantly lower percentage with only 13% of open articles or 5% of all articles. Diamond clearly is not commonly used by New Zealand researchers.

Table 3 Proportion of articles with at least one New Zealand author which were open.

Availability of article	Count	%	
Closed	7,049	59%	
Open	4,944	41%	
Total	11,993	100%	

Table 4 Proportion of articles which were open with a New Zealand university researcher listed as corresponding author.

Availability of article	Count	%	
Closed	3,501	66%	
Open	1,798	34%	
Total	5,299	100%	

Table 5 Articles by type of access.

Using our definitions of types of access, the proportion of each type in our dataset is shown. This table shows all publications with at least one author affiliated to a New Zealand university.

Type of access	Count	%	
Bronze	1,089	9%	
Closed	7,049	59%	
Diamond	265	2%	
Gold	1,706	14%	
Green	1,256	10%	
Hybrid	628	5%	
Total	11,993	100%	

It is worth noting the ‘overlap’ of publications available in a published OA form with repository versions. As detailed in our materials and methods section, we categorise types of OA using the Unpaywall method, which assigns a single type of OA to each publication and favours the published version over a Green one. While we report the proportion of Green-only OA in our sample as 10% or 1256 publications, we also investigated the number of publications that are available in Green and one of the other open forms: 3319 of 11993 (28%) total papers were available via a reputable repository. 2172 articles were available openly via the publisher and also via a repository, representing a considerable overlap. Of particular interest is the Bronze category, since these publications are of uncertain status and could revert to Closed. We found that 288 of 1089 Bronze papers in Table 5 were also available via a repository, meaning those would remain open even if the published version became Closed. If we add those 288 to our Green-only subset then the Green proportion would increase from 10% to 13% and Bronze would reduce from 9% to 7%.

We were also interested to determine how many publications were only available via a New Zealand university repository. Our dataset provides the URL for the repository version and by filtering for the string ‘.ac.nz’ (common to all eight New Zealand university repositories) we found that a total of 125 of the 1256 Green publications (10%) were only open because of that deposit. In other words the other 90% were available either as a published open version or in a non-New Zealand institutional or discipline-specific repository.

Again we analysed the subset of articles where a New Zealand university researcher was the corresponding author for the article (see Table 6).

The pattern is broadly similar to the dataset for all authors (as seen in Table 5). Gold is most common (39% of the open subset and 13% of all articles); Green and Bronze are near-equal at 24% and 23% respectively of the open articles and 8% of all articles; Hybrid is somewhat lower here, with 8% of open and just 3% of the total. Diamond is constant at 2% of all articles.

Table 6 Articles by type of access with a New Zealand corresponding author.

Using our definitions of types of access, the proportion of each type in our dataset is shown. This table shows all publications where the corresponding author is affiliated to a New Zealand university.

Type of access	Count	%	
Bronze	422	8%	
Closed	3,501	66%	
Diamond	95	2%	
Gold	697	13%	
Green	432	8%	
Hybrid	152	3%	
Total	5,299	100%	

Figure 2 Crossref citations for different types of access.

This boxplot compares the distribution of citations for each type of access. Citations, drawn from Crossref, are indicated on the Y axis. Average citation rates for each type are indicated by the horizontal line in each box. Each category had a number of outliers (depicted as dots). The graph is cropped at 20 citations on the Y axis to focus on the groupings of citations in the 0–10 range.

Citation rates

We examined the publications in our dataset for the number of citations reported by Crossref and broke these down by the different types of access.

Each category of access contains a large number of outliers, that is, publications that were cited well above average for each access category. Also of note, publications with no citations at all were also very high for Diamond (47%), compared with Bronze (34%), Closed (25%)—each of these sitting on zero of the Y axis in our boxplot—and with the remaining categories closely grouped (Hybrid 20%, and Gold and Green together on 18%). In general, Fig. 2 suggests a slightly higher rate of citation for open types of access compared to Closed, though one type—Diamond—performs the lowest in terms of citations. Table 7, which provides the standard error and the confidence interval of the mean for each category, reinforces Diamond as an outlier and higher rates of citation for the Green and Hybrid.

Table 7 Standard error and confidence interval for citation rates by access type.

Citations for each type of access, showing mean, median, standard deviation, standard errorand confidence interval for 95% confidence level.

	Count	Mean citations	Median citations	Standard deviation	Standard error	Confidence interval (95%)	
Bronze	1089	5.16	2	11.27	0.34	4.49, 5.83	
Closed	7049	4.53	2	9.51	0.11	4.31, 4.75	
Diamond	265	1.79	1	3.57	0.22	1.36, 2.22	
Gold	1706	5.14	3	7.75	0.19	4.77, 5.5	
Green	1256	7.54	3	26.23	0.74	6.09, 8.99	
Hybrid	628	7.50	3	15.53	0.62	6.29, 8.72	

Gold and Hybrid costs

We extrapolated the total number of articles that might have incurred an APC by adding together Hybrid and Gold figures. We see that 697 Gold articles and 152 Hybrid ones were published in 2017 in our local author subset (849 in total). The Program included a calculation of APCs for each article, where this was known via publicly-available data sources. This was calculated only for New Zealand university-affiliated authors on the basis that the corresponding author is the most likely to be responsible for paying an APC. While this is a far from certain means of determining how an APC was paid, as discussed by Gumpenberger, Hölbling & Ignacio Gorraiz (2018), it is the best one available from our sources and at scale gives the best approximation possible.

Thus Table 8 shows the average APC costs, US$2558 for Hybrid and US$1682 for Gold.

We were also able to estimate the total APCs paid. Most publishers provide information on publishing charges and this data has been collected by Lisa Matthias of the Freie Universität Berlin (Matthias, 2018). The ‘Known APC cost’ is a notional amount because:

• it is not possible to know where APCs may have been waived or whether they were paid from research funding, institutional funds, researchers’ own money or another source; and

• this information is not available for all journals.

The APC costs in our tables are effectively a total of the ‘list price’ for each article based on APC information that is publicly-available. Accordingly, the total amount for both categories was US$1.45 million at 2017 prices.

Table 8 Gold and Hybrid articles.

The table breaks down our Gold and Hybrid articles with a New Zealand corresponding author. Where known, the total Article Processing Charges and average changes for each type are calculated.

Type of paid OA	Count	%	Known APCs	Known APC cost	Known APC avg	
Gold	697	82%	697	$1,172,029	$1,682	
Hybrid	152	18%	110	$281,378	$2,558	
Total	849	100%	807	$1,453,407	$1,801	

Table 9 Articles that could be archived in a repository but were closed access.

Shows the proportion of closed articles broken down by embargo periods listed by publishers.

Embargo period	Publisher policy allows accepted manuscript in repository	%	
Immediate self-archive	579	17%	
3 months	3	0%	
4 months	1	0%	
6 months	73	2%	
12 months	2115	60%	
18 months	318	9%	
Total archivable by mid-2019	3089	88%	
24 months or more	213	6%	
Not self-archivable	199	6%	
Total closed articles	3501	100%	

Embargo periods and self-archiving

Sherpa/Romeo data let us examine which of the Closed articles could be self-archived according to publishers’ policies. Table 9 shows, for all New Zealand-corresponding authors, when a Closed article may be deposited in an institutional repository after an embargo set by the publisher.

We ran the Program in mid-2019, meaning any embargo period of 18 months or less would have expired. 3089 articles could have been archived but were Closed, representing 88% of all the Closed articles (n = 3501) in our sample set. If all of these were deposited then the overall open proportion would catapult from 41% to 67%. A further 212 articles have an embargo period of two years or more. It is worth noting that 12 months is by far the most common length of embargo period but also that for almost one-fifth there is no embargo. As we have noted above in the ‘Articles by type of access’ section, deposit in a New Zealand university repository only made the difference between open and Closed for 125 publications.

As a result we were also able to estimate a ‘theoretical’ cost of APCs under the Hybrid option for papers that could have been made open as accepted manuscripts. The total comes to just under US$8 million.

Also of interest is that for 114 of the 3089 articles that could have been deposited in a repository (3.7%) the publisher allowed the published version to be used, as opposed to the accepted manuscript.

The above calculations only consider publications where the corresponding author was from a New Zealand university, since the corresponding author is the most likely to have an accepted manuscript for deposit. However, if we extend our examination of Closed articles that could be deposited to all corresponding authors, again 88% have passed the embargo expiry date, which would mean 6,204 of all the 7,056 Closed papers could be placed in a repository. This would mean 93% of 2017 publications would be freely accessible without payment.

Table 10 Proportion of open articles funded by major New Zealand funders.

This table includes the New Zealand funders most represented in our dataset, showing the proportion of publications for each that are closed or open by the different methods of doing so. The table is broken into three: it shows the results for all authors and then splits these overall figures into two subsets, where the corresponding author was a researcher affiliated with a New Zealand university and where not.

Funder	Number	Closed	Bronze	Gold	Diam’d	Hybrid	Green	
All Authors								
Marsden Fund	505	54%	10%	13%	1%	5%	18%	
Rutherford Discovery Fellowship	90	44%	9%	21%	13%	12%	12%	
Royal Society of New Zealand	714	54%	9%	14%	1%	6%	16%	
Health Research Council of New Zealand	468	45%	13%	29%	1%	3%	9%	
Ministry of Business Innovation and Employment	443	65%	7%	16%	1%	5%	7%	
Total	1,519	55%	9%	19%	1%	5%	11%	
NZ Corresponding author								
Marsden Fund	362	57%	10%	13%	1%	4%	15%	
Rutherford Discovery Fellowship	68	51%	12%	13%	0%	12%	12%	
Royal Society of New Zealand	515	57%	9%	14%	1%	5%	14%	
Health Research Council of New Zealand	356	46%	13%	29%	1%	3%	8%	
Ministry of Business Innovation and Employment	303	65%	9%	17%	1%	3%	6%	
Total	1,098	56%	10%	19%	1%	4%	10%	
Non-NZ Corresponding author								
Marsden Fund	143	45%	9%	12%	1%	7%	25%	
Rutherford Discovery Fellowship	22	23%	0%	45%	0%	18%	14%	
Royal Society of New Zealand	199	47%	8%	15%	1%	9%	21%	
Health Research Council of New Zealand	112	43%	14%	27%	0%	4%	13%	
Ministry of Business Innovation and Employment	140	64%	4%	14%	1%	9%	8%	
Total	421	51%	8%	18%	1%	8%	15%	

Articles funded by New Zealand’s major funding agencies

Funder information from Web of Science and Scopus enabled us to estimate how much research funded by our major funding agencies is openly available, as shown in Tables 9 and 10. As indicated in our section on the context for the study, there has been no attempt by the government or major funding agencies to adopt a co-ordinated approach to open access in universities or to provide dedicated funds to support the payment of APCs. Similarly, none of these agencies release public information about outputs funded by them or the way in which they have been published.

More than half of articles in our 2017 sample that were funded by our largest research funders are behind a paywall (55%)—that is, this research is inaccessible to the government agencies that funded it as well as to the New Zealand public. Therefore, this subset of articles was more likely to be open than the total sample (45% open against 41% open). The Ministry for Business, Innovation and Employment has the lowest rate of open research at 35%, while the Health Research Council of New Zealand had the highest proportion with 55% of papers open. Papers with New Zealand corresponding authors and supported by these New Zealand funders were less likely to be open (44%) than those without New Zealand corresponding authors (49%). Detailed figures are presented in Table 10.

We can also see in Table 10 how the funded articles that are freely accessible have been made open. Gold was by far the most common means of making a work open (19% of all articles or 41% of all open publications); Hybrid accounted for just 5% of all articles or 11% of all open. Green made up 11% of all works or 25% of open and Diamond just 2%. Bronze means, by definition, that the permanence of the remaining 9% of all articles or 21% of open works is uncertain. Combining the figures for Gold and Hybrid where the corresponding author was a New Zealand researcher, 22% of freely accessible research funded by these agencies theoretically incurred an APC. We calculated these 249 articles to cost US$455,000 if the ‘list price’ was paid in each instance. This 22% compares to 14% of publications that would have incurred an APC where there was no funding from one of the major New Zealand government agencies. In other words, where work was specifically funded by one of these agencies an APC was more likely to have been paid.

Our sample set was large enough to allow comparisons with publications supported by public funding from Australia, the United States and the United Kingdom which showed substantially higher proportions of open publications for both New Zealand and non-New Zealand corresponding authors, as presented in Table 11.

Discussion

We found that three out of five articles with an author from a New Zealand university were only available by paying for access (59%). This figure increases to nearly two-thirds of all articles being closed when the corresponding author is a New Zealand university researcher (66%).

For validation of our results we looked at the Leiden ranking measure for openness. The Leiden Ranking (Centre for Science and Technology Studies, 2018) uses a different method to ours, including using data from 2014-17 and including only 5 of the 8 New Zealand universities, but produces a similar result (see Table 12).

We also used the Leiden Ranking tool to measure New Zealand’s proportion of open articles against a selection of other countries. We clearly see that New Zealand’s proportion of research that is openly available is below that of all the others in this selection, nearly half the figure of the highest-ranked nation, the United Kingdom. This is reinforced by our analysis of publications funded by New Zealand’s major funding agencies, where we can very clearly see that research funded by agencies in other countries is far more likely to be openly accessible.

Table 11 Comparison of funded articles by country.

This table shows the breakdown of access type to publications listed as funded by government funding agencies in New Zealand, the United States, Australia and the United Kingdom, showing the proportion of publications for each that are closed or open by the different methods of doing so. The table is broken into three: it shows the results for all authors and then splits these overall figures into two subsets, where the corresponding author was a researcher affiliated with a New Zealand university and where not.

Funder	No.	Closed	Bronze	Gold	Diam’d	Hybrid	Green	
All Authors								
NZ Govt	1,519	55%	9%	19%	1%	5%	11%	
US Govt	271	24%	18%	20%	2%	14%	22%	
Aust Govt	358	39%	11%	23%	1%	6%	20%	
UK Govt	199	12%	17%	27%	1%	20%	24%	
NZ Corresponding Authors								
NZ Govt	1,098	56%	10%	19%	1%	4%	10%	
US Govt	52	33%	12%	17%	4%	13%	21%	
Aust Govt	68	46%	10%	22%	0%	4%	18%	
UK Govt	33	33%	15%	15%	3%	12%	21%	
Non-NZ Corresponding Authors							
NZ Govt	421	51%	8%	18%	1%	8%	15%	
US Govt	219	22%	20%	20%	2%	15%	22%	
Aust Govt	290	37%	11%	23%	1%	7%	21%	
UK Govt	166	8%	17%	29%	0%	22%	25%	

A huge proportion (88%) of the Closed articles could be self-archived in line with publishers’ policies and thereby made open. Our findings suggest that New Zealand researchers do not self-archive as often as researchers elsewhere and/or that the systems for ensuring work is archived are not effective. This is despite the fact that 87% of New Zealand researchers believe that, at a policy level, publicly-funded research should be free to access (Ithaka, 2018). Our work identifies a clear gap between belief and practice.

When it comes to paid open access (Gold & Hybrid articles), New Zealand researchers are far more likely to use the Gold route (82% of paid open access articles were Gold). One reason for this may be the higher average APC for Hybrid, which may be seen by researchers as a luxury and opted for when publishing work in a prestigious journal that will garner interest within the discipline and/or from the public. This would require further analysis that was beyond the scope of the present project. We estimated a total of US$1.45 million at 2017 prices could have been spent by our researchers: US$1.17 million of this would have been an entirely additional cost to subscriptions for Gold OA publications; US$281 thousand would have been on Hybrid publications, a potential double-dipping cost on top of subscriptions to those same publications. As we mention in our introduction, discussion of transformative agreements with publishers is outside the scope of our paper but our data can certainly be used by libraries and institutions to estimate the costs that are potentially being paid by researchers on top of subscription costs.

Table 12 Leiden ranking proportion of open articles by country.

Using Leiden Ranking data, this table lists the number of publications in eight countries andthe proportion of articles that are open access.

Country	# Papers	# OA papers	Percentage OA	
UK	454,802	322,827	71.0%	
Norway	42,608	23,109	54.2%	
US	1,876,219	1,013,502	54.0%	
Ireland	26,548	12,966	48.8%	
Germany	397,439	190,543	47.9%	
Canada	281,304	117,247	41.7%	
Australia	273,486	113,789	41.6%	
New Zealand	29,091	11,266	38.7%	

For our methodology, using the Unpaywall categorisation of openness means Bronze articles pose something of a quandary. Bronze was introduced by Unpaywall to be able to include papers openly accessible at a given point in time, but lacking definitive licensing information. With our Program this meant, however, that later iterations using the same DOIs (not reported on in the present paper) revealed that many papers categorised as Bronze in May 2019 had reverted to Closed or had switched to Green. The Unpaywall hierarchy places Bronze above Green, since it is the published version, but there is no way of knowing which papers will become Closed if publisher paywall restrictions are reimposed and which will continue to remain accessible through repositories. Fortunately, because Unpaywall provides repository locations in addition to the primary status it is possible to identify these Bronze/Green articles, which, for our sample, constituted 26% of all Bronze Papers.

In terms of citations, the complicated nature of citation advantage (or disadvantage) is well documented in the literature (Gaulé & Maystre, 2011; Mikki, Gjesdal & Strømme, 2018; Torres-Salinas, Robinson-Garcia & Moed, 2019). As noted in our review section, research largely suggests a correlation between openness and higher citation rates, though this is difficult to quantify, given contributing factors like disciplinary differences, the choice of publication venue by researchers, or the means by which citation rates are calculated. From our analysis, it is difficult to say definitively that open access confers a citation advantage, since different approaches to the question yield different answers. Given that two different types of open perform at opposite ends of the citation spectrum—Hybrid and Diamond—this seems to support the view that consideration needs to be given to factors such as journal choice by researchers or disciplinary norms for citation rates, which are outside the scope of the current study.

Nevertheless, our work does seem to support previous findings that there is a positive correlation between open access and higher citations (Archambault et al., 2014; Copiello, 2019; McCabe & Snyder, 2014; Mikki, Gjesdal & Strømme, 2018; Ottaviani, 2016; Piwowar et al., 2018; Piwowar, Priem & Orr, 2019; Wang et al., 2015), though we note the number of outliers in each category that are highly-cited. One interesting subset for comparison is Closed and Green access, since Green publications are those that would be Closed but for an automated deposit process or a conscious decision by a researcher or institution to make that work open. Our results suggest slightly higher rates of citation for Hybrid and Green; Closed does appear to perform slightly below other forms of access, with the exception of Diamond.

Also of note, articles that listed a major New Zealand funding agency achieved a higher overall rate of openness than the whole sample set (45% as opposed to 41%). However, this is still low considering such projects are funded specifically because they are deemed to be socially or economically valuable research to pursue and therefore worthy of targeted public funding. It should also be noted that there is a good deal of variance within the individual agencies (as low as 35% to as high as 55% open), which again evidences the lack of co-ordination amongst funders, including the government, in New Zealand. We can also see that Gold and Hybrid account for 22% of papers with a New Zealand corresponding author and that it is more likely that researchers with this kind of funding publish by paying an APC.

As we have seen, 3,089 articles that were Closed could have been deposited in a repository. This number will have increased in the time that has elapsed since we conducted our analysis and the publication of this paper, since 24-month embargoes will have also expired. This represents an interesting consideration for universities. Clearly our institutional repositories are under-utilised if only 125 publications from 2017 are the only open version available. If those 3,089 Closed articles were deposited then the overall proportion of open would catapult from 41% to 67% with the Green contribution increasing from 10% to 36%. A 2015 study found that the processing cost of depositing an article in an institutional repository, including the time of the author, was £33 (or about US$43) (Johnson, Pinfield & Fosci, 2016). Using this figure the 3,089 articles that are closed but could be open would cost US$132,870. This compares to the US$1.45 million identified in our project as potentially paid in Gold and Hybrid APCs and the amount reported by CONZUL as spent in 2017 by university libraries on subscription to electronic resources, NZ$68.5 million (around US$45 million) (Universities New Zealand, 2019).

Limitations of this research

We reiterate that the programmatic nature of our method means this does not represent all research, only articles with a DOI. Thus there will be disciplinary skews to the sample set, since journal articles and DOIs are more prevalent in certain disciplines. The research could easily be expanded to incorporate book chapters or other types of work that have a DOI. Nevertheless, not all research falls within the scope of our analysis.

As we have noted in our discussion above, Unpaywall, upon which much of our data gathering depends, updates its database constantly, including the repositories it sources information from. Thus any time the Program is run the results depend on the current state of the Unpaywall database. This can result in fluctuations in results even using the same set of DOIs when the Program is run at different times. Bronze access articles may, by their nature, change status over time. This is not, in itself, problematic, but is noted here only because any set of data produced by the Program is a snapshot of a moment in time. We do intend to re-run the Program and do an analysis each year to track trends over time.

It is also possible that other open access discovery APIs could be utilised, such as the CORE API (https://core.ac.uk/services/api/). Use of different databases would naturally lead to different results, though we do not anticipate this would affect the overall findings significantly for New Zealand-based articles.

Another limitation is that our calculations of the amount spent on APCs is a maximum amount based on published prices, as noted in the section on our findings. Actual amounts paid will almost certainly be less because there will have been waivers or discounts applied.

We also note that deeper investigation of citation rates for different types of access would be of interest, building on our findings by calculating normalised citation rates and comparing differences between disciplines.

Finally, with respect to estimates of research funded by New Zealand’s funding agencies, we noted above that those agencies do not provide publicly-available lists of research outputs they have funded and the means of publication. Thus there is no way for us to verify funder information reported by Web of Science and Scopus.

Conclusions

In May 2019 we ran our specially-developed software to discover that about two out of every five articles authored by New Zealand researchers in 2017 were freely available on the web (41%). This is the first time we have an evidence-based picture of access to research by New Zealand universities with such detail since, as a result of our work, we have far more than a simple overall proportion: we can investigate the ways in which work has been made accessible, we can compare the citation rates for these different modes of access, we can quantify the volume of works that are closed access and could be made open and we can estimate how much paid forms of open access have cost.

Since our code is publicly available, anyone can run their own set of DOIs to perform their own analysis of these aspects.

Overall, we see that more New Zealand research from 2017 is behind a paywall than is freely accessible (41% freely accessible, 59% closed). However, when the corresponding author was a New Zealand researcher the open figure drops to around a third (34%). When our major funding agencies have specifically funded the research the proportion of articles that is accessible is higher but still just under half is accessible without a subscription.

Where work is freely accessible, Gold is the most likely means of achieving this at an average cost of USD1,682 per article; while Hybrid is used significantly less often it comes with a higher average price (USD2,558). In all the two paid methods of making research accessible comes with an estimated price tag—on top of library subscription costs, of course—of USD1.45m.

Green open access accounted for about one-quarter of our open articles. One further avenue we can investigate is where this work was archived, whether in our own university repositories or in public ones like PubMed. We found that this proportion could be greatly increased if our authors utilised the rights afforded to them by publishers to make versions of their work freely accessible in non-commercial repositories. Fully 3,089 (88%) of Closed articles could be made available in this way but in our 2017 sample we identified a paltry 125 articles in New Zealand’s institutional repositories that made the difference between open and closed.

These findings beg several questions worthy of further research that are outside the scope of this paper. What are the barriers to self-archiving? The most likely reasons—of which we are aware from our own anecdotal experiences—are lack of time, lack of awareness of the possibility of self-archiving, confusion about copyright and embargo periods, negative perceptions of the status of author accepted manuscripts, and the lack of user-friendliness of software used to deposit works in a repository. Why do our researchers choose one mode of publication over another? Which publishers do our researchers favour when choosing open? What influences them to choose to pay a Hybrid APC? Does journal impact factor play a role in decisions or in citation rates? Are there disciplinary differences?

Our data can also be interrogated further than was possible within the scope of this paper. While our focus was on a national snapshot of open and closed publications, it is also possible for individual institutions to examine their own subsets of our data to determine costs and identify closed publications that could be deposited in their repositories. We have already mentioned that information garnered about APC payments can help institutions to estimate amounts paid by researchers for publishing when looking at the value of transformative publishing agreements with publishers.

What we do know is that New Zealand research is less likely to be open than research of other countries. Our overall proportion of open work lags behind other countries, our corresponding authors are less likely to make research open than corresponding authors from other countries and, clearly, we could be taking advantage of Green open access to a far greater extent than we are. This last point in particular suggests there are important policy and systemic issues that should be considered by New Zealand’s research community. Despite the fact we know most authors support open access to research in principle, there is a very large gap between this belief and their practices in making New Zealand’s research outputs free to access.

Supplemental Information

Supplemental Information 1 New Zealand university-affiliated journal articles published in 2017 (raw data, analysis and research notes)

Click here for additional data file.

This research formed one part of a larger project under the auspices of the Council of New Zealand University Librarians. The authors acknowledge CONZUL’s guidance and support.

Additional Information and Declarations

Competing Interests

Author Contributions

Data Availability

All authors have roles within their organisations that may involve advising colleagues on open access issues, which may include advocacy for benefits associated with open access. Luqman Hayes is a member of the Tohatoha Aotearoa Commons Council.

Richard K.A. White analyzed the data, prepared figures and/or tables, authored or reviewed drafts of the paper, and approved the final draft.

Anton Angelo analyzed the data, authored or reviewed drafts of the paper, reviewed software code, and approved the final draft.

Deborah Fitchett, Moira Fraser, Luqman Hayes, Jessica Howie and Emma Richardson analyzed the data, authored or reviewed drafts of the paper, and approved the final draft.

Bruce White conceived and designed the experiments, performed the experiments, analyzed the data, prepared figures and/or tables, authored or reviewed drafts of the paper, author of software code, and approved the final draft.

The following information was supplied regarding data availability:

The software to perform analysis on a set of DOIs, a Python program to harvest metadata and metrics from DOIs of scholarly publications are available at GitHub: https://github.com/bruce-white-mass/conzul-oa-project.

The raw data used in our analysis is available in the Supplemental File.

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
