# Peer review of "Only two out of five articles by New Zealand researchers are free-to-access: a multiple API study of access, citations, cost of Article Processing Charges (APC), and the potential to increase the proportion of open access"

_PeerJ, doi:10.7717/peerj.11417_

## Round 0.1 · original submission · Major Revisions

Both referees have done a very thorough job in their review including trying to replicate some of your results. Both found some differences between the results in the paper and what they found though the differences aren't major. Please provided a detailed response to the referee comments together with your revised submission.

·

Basic reporting

The manuscript of White et al is a clear and well constructed description of the state of Open Access in New Zealand for articles published in 2017.

The literature review covers the main topics well and discusses issues with the measurement of open access levels. The authors may want to consider updating the literature with two recent pieces of work (Robinson-Garcia N, Costas R, van Leeuwen TN. 2020. PeerJ 8:e9410 https://doi.org/10.7717/peerj.9410 and Huang et al 2020.BiorXiv https://doi.org/10.1101/2020.03.19.998336, acknowledging that our recent piece is a preprint at this point, currently undergoing peer review).

The tables are clear and well laid out and the conclusions are justified by the evidence presented. I note that the data, and the code is made available. I have some queries below with respect to data analysis and provide my own attempts at replication.

The replication is conducted in a Jupyter notebook which is available at:
https://github.com/cameronneylon/review_white_2020

and as a PDF via attachment to this report.

Experimental design

The experimental design is well thought through and described with some caveats on the clarity of the analysis procedure. The data is provided and code is also provided that generates the output data.

I have not attempted to run the code as I do not have a Sherpa Romeo api key. I have however conducted some elements of a conceptual replication and find the results consistent (below).

I have two questions for clarification on the details of the analysis. These are based on my replication of the reported analysis using the dataset provided.

Details of the analysis and filtering

The provided datafile has 12,931 lines, not 12,600 as described in the article text. In addition when I filter down to articles with 20 or fewer authors I get 12,226 not 12,016 articles. Similarly there are small differences in the numbers throughout my analysis.

I would recommend the addition of either a flow-chart which explicitly walks through the filtering process (including precise numbers of entries at each point) and how the provided datafile was generated and its relationship to the code. Clearly I took a slightly different process, and broadly speaking the numbers I obtain in my replication from the provided data are very similar, but not quite identical. A fully replicable process would enhance the confidence in the conclusions.

Analysis by funder

I have some concerns or misunderstandings about the analysis by funder. In my own analysis of the provided dataset I noted a lot of variant names for funders, some of which appear as if they are variants of the names used (see replication report). For instance Health Research Council, Health Research Council of New Zealand, and HRC all appear, as well as variants of the Ministry of Business Innovation and Employment and the Royal Society of NZ.

In my analysis the numbers for the precise string noted in Table 9 was usually the closest but this suggests a significant under counting of funded research. Interestingly there seem to be significant variations in the proportion of OA under different names, which might provide some help in estimating error margins?

It may also be of value to analyse some of the non-New Zealand funders that appear and potentially differences within funders for the case where there is a non-New Zealand corresponding author. It would be interesting, for instance if the levels of OA for the same funders were higher where the corresponding author is not from New Zealand.

Confirmation via conceptual replication using alternate datasources

In the provided replication report I have also conducted a conceptual replication for the overall numbers and levels of OA and for funders. I conduct this by querying a local copy of Microsoft Academic Graph for the GRID IDs of the eight New Zealand Universities, identifying a little over 9,000 outputs for 2017. Overall I find similar percentages for open access and for average citation counts based on MAG data which offers solid support for the main findings here.

I did not have ready access to SHERPA ROMEO data or to APC data, nor did I have corresponding author data so did not attempt a conceptual replication of those analyses.

Validity of the findings

The overall conclusions are supported by the evidence and both the direct replication and the conceptual replication (where conducted) support the overall findings.

As noted above I recommend a more detailed description of the data handling, filtering and precise details of the production of the provided dataset. I also recommend checking over the funder data.

Additional comments

Thank you for this work. As noted above my two main comments relate to:

1. The details of the methodology and production and filtering of the data.
2. The analysis by funder and possibility that some funders may have been missed in the analysis

On the content itself I wonder whether it is worth considering the degree to which repository mediated or green oa is contributing overall. In my conceptual replication I note that 29% of the content is available via green OA which is close to the 37% of total OA from my data. This to my mind strengthens the argument for green as the most direct route to increasing overall levels of open access.

This is not obvious from the current text because a proportion of this is labeled as gold. Given that those authors who are engaging with OA are clearly using repositories of one type or another for both closed content but also hybrid and gold, this supports the case that if more authors can be engaged then the green route is the most cost effective forward.

Further, I would argue for a change in terminology to make this explicit, noting that 'gold' here is "in an a pure oa journal" to distinguish between uses that combine this with hybrid, and using "green only" or some other term to make explicit that this applies to repository-mediated OA for outputs that are also published OA. It might help to actually add those two separate categories of green to the tables.

·

Basic reporting

The article is generally well written in professional language. The structure of the article adheres to the PeerJ guidelines. The background acknowledges recent approaches to analyse open access in terms of productivity and citation advantage. Tables presents summary statistics; no figures are provided. The supplement contains article-level data. The authors share the source code of a tool developed for this study via GitHub, which is great!

I suggest to improve introduction and background by clarifying some open access terminology.

Most importantly, Peter Suber's seminal work on open access (OA) (MIT Press, 2012) is not acknowledged according to which Gold OA is defined as "OA through journals, regardless of the journal’s business model." In the article, however, Gold OA refers to fully open access under an APC business model (lines 63-64). Please discuss why you restrict the term "Gold OA" to journals with a specific OA business model, or revise the terminology. Similarly, "Green open access" is introduced as open access through institutional repositories (lines 67-68), but there are also other relevant types of repositories like subject-specific repositories. Is the focus of this work just on institutional repositories?

Because the APC business model plays a central role in the article, I furthermore suggest to acknowledge recent discussions about transitioning (library) spending for journal subscriptions to open access. In particular, the influential White Paper from the Max Planck Society from 2015 (https://doi.org/10.17617/1.3) is relevant as well as related work from the US-American Pay it Forward (https://www.library.ucdavis.edu/icis/uc-pay-it-forward-project/) project. I also suggest to review the work from Gumpenberger et al. (2018, https://doi.org/10.3389/frma.2018.00001) where the authors discuss bibliometric issues using corresponding authorships in the context of open access funding.

In addition to the Leiden Ranking, I suggest to consider the recent preprint "Evaluating institutional open access performance: Methodology, challenges and assessment" from Huang et al. (2020, https://doi.org/10.1101/2020.03.19.998336). In this article, the authors examined and discussed evidence sources to measure the OA performance of universities.

Experimental design

The methods could be described in greater detail. Particularly, it is not clear to me when and how the underlying article DOIs were gathered. Did they originate from searching the Web of Science and Scopus? If so, what search queries were used? What was the overlap between these two sources relative to the New Zealand research output? And what was the proportion of articles without DOIs?

Research questions are descriptive, seeking to quantify the open access performance of New Zealand's university. I think the supplemented data is very comprehensive and would allow to ask for variations by subject or publisher. It would be also great to learn more about whether and why the OA uptake varies across the universities.

Validity of the findings

The authors supplemented article-level data. Unfortunately, I did not succeed replicating the presented results using the spreadsheet, but obtained slightly different numbers. Please find my R Markdown notebook attached to this review. Please also provide the underlying code used to generate the tables.

Besides, I am missing important context to assess the validity of the findings. Articles with more than 20 authors were excluded because they had a "tenuous connections" to the studied universities. Why did you choose the threshold of 20 authors? And how did you validate the "tenuous connection"? Did you consider alternative counting methods like fractional author counting?

I also miss background about how the accuracy of the retrieved article-level data was assessed in terms of recall and precision. How did you assure the quality of the tool and the open access evidence sources?

Drawing the conclusion that open access has a citation advantage based on the presented methods and results is very problematic. I feel that the citation window between the publication year of 2017 and mid 2019, the date of gathering the citation data, is too narrow to draw any meaningful conclusion. Furthermore, only mean citation rates were reported. The citation rates were also not normalized for differences in citation patterns between research fields, which is a standard practise in bibliometrics.

Additional comments

I would suggest that you narrow the focus of the study. The data you collected so far are great and provide more analytical opportunity than what was presented. One promising focus is the debate around transformative agreements and the economic consequences for library budgets and national library consortia in the future. Another promising approach is to highlight variations between universities and the exploration of potential factors affecting the institutional OA uptake. If the focus is on the tooling, then the paper should be revised to reflect the technical focus.

---

## Round 0.2 · Major Revisions

The first reviewer is now happy with your paper and just requires a few issues being fixed. Please address these. The second reviewer has concerns about the citation analysis and attaches his interpretation of the results. Please address these issues.

·

Basic reporting

The authors have addressed all of the raised issues with regard to clarity and some issues of confusion. I will note for the record that I disagree with some of the terminological choices but given there is no widely accepted standard for defining types of open access I do not think this is worth changing. The definitions are given clearly enough in the article.

Minor issue: Our preprint, referenced in the article at line 714 is now published at eLife - https://doi.org/10.7554/eLife.57067

Experimental design

The results are well defined and the process is now much clearer.

Minor issues: I can completely replicate the results given from the provided data. However there is one slight issues. There are 21 DOIs for which no authors appear to have been recovered and these are interpreted as having less than 21 authors. If this is correct then it would be valuable to make that clear in Figure 1.

Validity of the findings

I find the overall findings robust and clear. I note that I have conducted an outline conceptual replication of some aspects of the paper using an independent dataset and confirm the general findings.

https://github.com/cameronneylon/review_white_2020

There are variations in both the set of DOIs detected in the replication and the details of the results, particularly that the citation counts have subsequently changed following the data capture for the paper, but overall the conclusions are consistent with those drawn in the article.

Additional comments

Thank you for this article and the work done on it. I look forward to seeing the response to it.

·

Basic reporting

No comments

Experimental design

No comments

Validity of the findings

The revision did not sufficiently addressed my concern regarding the citation analysis. By just comparing mean citation counts, neither a citation advantage for open access nor for OA types can be derived. To illustrate my concerns, I used the provided spreadsheet and compared the OA distributions (for replication, my R Markdown notebook output is attached using the "annotated PDF file" functionality"). My exploratory analysis suggests that the citation distributions closed vs OA as well as by OA type are quite similar with a few outliers. The exemption is diamond OA, but this OA types accounts for smallest number of publications in the sample. Moreover, at least two highly cited and highly collaborative OA publications were miscoded in the spreadsheet as having 20 or less authors, and should have been excluded according to the study design.

Again, I strongly suggest to revise the citation analysis and provide a statistically valid proof for an OA citation advantage. Data errors need to be fixed.

Additional comments

Thank you for the revision and the nice job addressing most of my comments. However, I think that the citation analysis cannot prove a citation advantage, and should be thus substantially revised or omitted.

---

## Round 0.3 · Minor Revisions

Thank you for revising your paper and responding to the reviewer comments. The second reviewer was concerned that the differences between citations for the different access models were not statistically significant. It's good that you toned down the statement about citations, but I would like you to go further. I think it is fine to present the boxplot - this is informative - but I want more qualification of the "positive correlation". The sample is quite big, so it's possible that the differences in the means of the different access models are statistically significant. But we don't know without doing a test. On the other hand, we also don't know the direction of causation. Perhaps researchers put their best research into hybrid journals as open access, So, I would like you to at least write what I just wrote here. You have the following options:

1. Just write that you haven't tested for the statistical significance of these differences and that we don't know the direction of causation.

2. Alternatively also compute the standard error of the mean and add a confidence interval of the mean for each form of access to the figure or give it in the text. This would give people a rough idea of whether these differences are likely to be statistically significant or not. You could also write instead "this should be formally tested in future research".

3. Go the full way and do a test. You probably don't want to go as far as this.

If you make these changes adequately, I will accept the paper.

---

## Round 0.4 · accepted · Accept

Thank you for making the final changes to the article. This makes the comparison between types of OA much easier to understand.